# Using Blood Transcriptome Analysis to Determine the Changes in Immunity and Metabolism of Giant Pandas with Age

**DOI:** 10.3390/vetsci9120667

**Published:** 2022-11-30

**Authors:** Song Liu, Caiwu Li, Wenjun Yan, Senlong Jin, Kailu Wang, Chengdong Wang, Huiling Gong, Honglin Wu, Xue Fu, Linhua Deng, Changwei Lei, Ming He, Hongning Wang, Yanxi Cheng, Qian Wang, Shanshan Lin, Yan Huang, Desheng Li, Xin Yang

**Affiliations:** 1Key Laboratory of Bio-Resource and Eco-Environment of Ministry of Education, Animal Disease Prevention and Food Safety Key Laboratory of Sichuan Province, College of Life Sciences, Sichuan University, Chengdu 610017, China; 2China Conservation and Research Center for the Giant Panda, Key Laboratory of State Forestry and Grassland Administration on Conservation Biology of Rare Animals in the Giant Panda National Park, Chengdu 610083, China; 3Sichuan Wolong National Nature Reserve Administration, Wenchuan 623006, China

**Keywords:** time series, transcriptome, giant panda, immune system, aging, gene expression

## Abstract

**Simple Summary:**

Giant pandas are considered a national treasure in China. Understanding the changes in immunity and energy metabolism of giant pandas with age might help develop more scientific guidelines for managing the health of giant pandas. Here, 48 giant pandas were collected, and their transcriptome was analyzed. The results showed that the immune system and energy metabolism of giant pandas changed significantly with age.

**Abstract:**

A low reproductive rate coupled with human activities has endangered the giant panda, a species endemic to southwest China. Although giant pandas feed almost exclusively on bamboo, they retain carnivorous traits and suffer from carnivorous diseases. Additionally, their immune system is susceptible to aging, resulting in a reduced ability to respond to diseases. This study aimed to determine the genes and pathways expressed differentially with age in blood tissues. The differentially expressed genes in different age groups of giant pandas were identified by RNA-seq. The elderly giant pandas had many differentially expressed genes compared with the young group (3 years old), including 548 upregulated genes and 401 downregulated genes. Further, functional enrichment revealed that innate immune upregulation and adaptive immune downregulation were observed in the elderly giant pandas compared with the young giant pandas. Meanwhile, the immune genes in the elderly giant pandas changed considerably, including genes involved in innate immunity and adaptive immunity such as PLSCR1, CLEC7A, CCL5, CCR9, and EPAS1. Time series analysis found that giant pandas store glycogen by prioritizing fat metabolism at age 11, verifying changes in the immune system. The results reported in this study will provide a foundation for further research on disease prevention and the energy metabolism of giant pandas.

## 1. Introduction

The giant panda (*Ailuropoda melanoleuca*) is endemic to China. However, they are in danger of extinction due to habitat shrinkage as a result of the expansion of the human population [1,2]. The giant panda’s diet consists almost entirely of bamboo, but it retains the digestive tract characteristics of a carnivore, making it difficult for the giant panda to obtain energy from food. Recent studies have shown that giant pandas can facilitate bamboo digestion through a unique gut microbiome [3]. A mutation in the DUOX2 gene allows pandas to maintain low thyroid hormone levels to reduce energy expenditure. The smaller size of some organs (brain, liver, kidneys) also helps pandas conserve energy [4]. Studies in recent years have proved that the giant panda has a broad genetic diversity [5,6,7], contrary to the common opinion that the endangered giant panda is an evolutionary failure due to its small population and low fertility. However, most of these studies have focused only on panda diversity and adaptability.

Some viruses or parasites still trouble the giant panda as its digestive tract retains its carnivorous identity [8,9]. *Baylisascaris Schroederi* and canine distemper virus (CDV) are among the parasites and viruses responsible for causing numerous casualties, killing half of the panda population between 2001 and 2005 and five pandas between 2014 and 2015 [8,10].

The functioning of the immune system is closely related to age, with the body’s ability to respond to infection and the immune responses to vaccination decreasing with age [11], increasing the body’s susceptibility to various infections. Aging is an inevitable biological process, complicated by its interactions with the environment, genetics, and age-related diseases [12]. The gradual decline in the immune system with age has been demonstrated in several species [13,14,15]. Studies on the expression of age-related genes have shown that genes related to inflammation and immunity are overexpressed during aging, while those related to energy are suppressed [16]. Since the giant panda has a special status in China, understanding the changes in immune systems with age could help prevent age-related diseases.

This study aims to explore the changes involved in genes and pathways in different growth and developmental stages of giant pandas and to understand the changes in immunity-related genes. Age-related differentially expressed genes (DEGs) were identified in the giant panda blood samples using RNA-seq technology and analyzed for functional enrichment. The changes in the immune system and metabolism of giant pandas with age were studied, which showed a certain complementary role for the research of giant pandas.

## 2. Materials and Methods

### 2.1. Sample Collection

A total of 48 blood samples were collected, from 16 male and 32 female giant pandas, Wolong Nature Reserve, China. Giant panda blood was collected by professional personnel, and all blood was immediately loaded into BD PAX gene blood collection vessels. Each tube was loaded with about 2.5 mL of blood samples, which were mixed and stored at −80 °C.

### 2.2. RNA Acquisition in Blood Samples

After blood was removed from the refrigerator, it was placed at room temperature for at least 2 h. Then, 3000–5000× *g* was centrifuged for 10 min to remove the supernatant; 4 mL RNase free water was added, shaken with a vortex until the sediment was dispersed, centrifuged 3000–5000× *g* for 10 min, and the supernatant was removed; 350 μL BR1 was added, shaken and mixed well, 300 μL BR2 and 40 μL protease K were added, vortexed and mixed well for 5 S, incubated at 400~1400 rpm at 55 °C for 10 min; all the mixed solution was passed through a PAX gene Shredder spin column, centrifuged at full speed for 3 min, then the supernatant was transferred into a 1.5 mL centrifuge tube, 350 μL absolute ethanol was added, and vortexed evenly; The supernatant was transferred into a PAX gene RNA spin column and centrifuged at room temperature of 8000~20,000× *g* for 1 min; 350 μL BR3 was added, centrifuged at room temperature of 8000~20,000× *g* for 1 min; 70 μL DNA digestion buffer and 10 μL DNase I were added, and placed at 20–30 °C for 15 min; 350 μL BR3 was added, centrifuged at room temperature with 8000~20,000× *g* for 1 min, and the waste liquid was discarded; 500 μL BR4 was added, centrifuged at room temperature with 8000~20,000× *g* for 1 min, and the waste liquid was discarded; 500 μL BR4 was added, centrifuged at room temperature of 8000~20,000× *g* for 3 min, and the waste liquid was discarded; the purification column was transferred to a new 1.5 mL centrifuge tube, 20–50 μL BR5 was added, and centrifuged at room temperature with 8000~20,000× *g* for 1 min to elute RNA.

### 2.3. Library Preparation and Sequencing

1.mRNA Isolation

mRNA molecules were purified from total RNA using oligo(dT)-attached magnetic beads.

2.mRNA Fragment

mRNA molecules were fragmented into small pieces using fragmentation reagent after reaction a certain period in proper temperature.

3.First Strand cDNA Synthesis

An appropriate amount of primers was added to the interrupted sample, mixed well, and reacted at a suitable temperature on a thermal cycler for a certain period of time to open the secondary structure and combine with the primers. Then, the first-strand synthesis reaction system was added, and the mix was prepared in advance. The first-strand cDNA was synthesized according to the corresponding procedure on the instrument.

4.Second Strand cDNA Synthesis

A second-strand synthesis reaction system was prepared (using dUTP instead of dTTP), then reacted on Thermomixer at an appropriate temperature for a certain amount of time to synthesize second-strand cDNA. The reaction product was purified by magnetic beads.

5.End repair and Add ‘A’

Prepare the end repair & add “A” reaction system, react on a thermal cycler for a certain time, under the action of enzymes, repair the sticky ends of the cDNA double-stranded by reverse transcription, and add A base to the 3′ end.

6.Adaptor Ligation

Prepare the linker connection reaction system. React on a thermal cycler at a suitable temperature for a certain time. Under the action of the enzyme, connect the linker to the A base. The reaction product was purified by magnetic beads.

7.PCR

Prepare the PCR reaction system, digest the U-labeled second-strand template with UDG enzyme and perform PCR amplification. PCR products were purified with XP Beads, and dissolved in EB solution.

8.Library Quality Control

The library was validated on the Agilent Technologies 2100 bioanalyzer (Agilent, Santa Clara, CA, USA).

9.Circularization

The double stranded PCR products were heat denatured and circularized by the splint oligo sequence. The single strand circle DNA (ssCir DNA) were formatted as the final library.

10.Sequencing

The library was amplified with phi29 to make DNA nanoball (DNB) which had more than 300 copies of one molecular. The DNBs were load into the patterned nanoarray and single end 50 (pair end 100/150) bases reads were generated in the way of combinatorial Probe-Anchor Synthesis(cPAS).

### 2.4. Quality Control 

This project used the filtering software SOAPnuke (version: V1.4.0, The Beijing Genomics Institute, Guangdong, China), independently developed by UBC, for filtering. The specific steps were the following:(1)Remove reads containing joints (joint contamination);(2)Remove reads with unknown base N content greater than 5%;(3)Removal of low-quality reads (low-quality reads were defined as those with a mass value of fewer than 15 bases that accounted for more than 20% of the total number of bases in the reads).

The filtered “Clean Reads” were saved in FASTQ format.

### 2.5. Gene Enrichment Analysis

According to GO annotation results and official classification, differential genes were functionally classified, and phyper function in R software was used for enrichment analysis. According to KEGG annotation results and official classification, the differential genes were classified into biological pathways, and phyper function in R software was used for enrichment analysis. The *p*-value was then FDR corrected. Usually, a Q value ≤ 0.05 was considered a significant enrichment.

## 3. Results

### 3.1. Transcriptome Sequencing and Assembly

In this project, 48 samples were tested using the DNBSEQ platform. The age distribution of the samples ranged from 3 to 30 years. Table 1 shows the sample ages and the biological repetitions. Each sample yielded an average of 6.44 Gb of data. The average rates of genome alignment and gene set alignment were 91.05% and 82.71%, respectively. A total of 22,013 genes were detected, including 21,104 known genes and 909 predicted new genes. Out of the 30,362 new transcripts detected, 29,415 belonged to new alternative splicing subtypes of known protein-coding genes, and 947 belonged to new protein-coding genes.

### 3.2. Identification of Age-Related Differentially Expressed Genes

Giant pandas have a natural life span of about 30 years. Although pandas typically live for about 20 years in the wild, they can live up to 30 years of age in captivity [17]. With the three-year-old group as the control, the statistics of differentially expressed genes in different age groups are shown in Figure 1. The age 3 group, the age 11 group, the age 18–19 group, and the age 27–28–30 group were selected, with at least three biological replicates for each group to facilitate the analysis. Differential genes (DEG) were identified and analyzed using DEseq2 software [18]. The results showed that 151 genes were significantly different in the age 11 group compared to the age 3 group, 83 genes were upregulated, and 68 genes were downregulated. The expression of 111 genes was significantly different in the age 18–19 group, 43 of which were upregulated and 68 of which were upregulated. Further, compared with the age 3 group, 949 genes were significantly different in expression level in the age 27–28–30 group, of which 548 genes were upregulated, and 401 genes were downregulated (Figure 2A).

### 3.3. Gene Ontology Enrichment Analysis of Differentially Expressed Genes

GO enrichment analysis was performed on the DEGs to understand their biological role further. Analyzing the GO term, it was observed that compared to the age 3 group, the upregulated genes in the age 11 group were significantly enriched in response to another organism (GO:0051707), defense response (GO:0006952), and response to the virus (GO:0009615) and 78 GO terms (Figure 3, Appendix A). However, the downregulated genes were not significantly enriched in any of the GO terms. Compared with the age 3 group, the upregulated genes in the age 18–19 group were significantly enriched in only two GO terms, defense response to a virus (GO:0051607) and response to a virus (GO:0009615). The downregulated genes were enriched in only two GO terms, immunoglobulin complex (GO:0019814) and negative regulation of Lipoprotein lipase activity (GO:0051005). GO enrichment analysis of differentially differentiated genes in the age 27–28–30 group found that Immune System Process (GO:0002376), Immune Response (GO:0006955), defense Response (GO:0006952), and other 273 GO terms with significantly enriched upregulated genes (Figure 2C, Appendix A). The GO terms enriched by downregulated genes included lymphocyte activation (GO:0046649), B Cell receptor signaling Pathway (GO:0050853), lymphocyte differentiation (GO:0030098), and B cell activation (GO:0042113), and 49 others (Figure 2E, Appendix A).

### 3.4. Pathway Enrichment Analysis of Differentially Expressed Genes

KEGG pathway enrichment analysis was also performed to evaluate the biological significance of DEGs further. It was observed that upregulated genes were significantly enriched in cytosolic DNA sensing pathway (KO04623) in the age 11 group. The downregulated genes were not significantly enriched in any pathway. In the age 18–19 group, the upregulated genes were not significantly enriched in any pathway, but the downregulated genes were significantly enriched in the B Cell receptor signaling pathway (KO04662) and fat digestion and absorption pathway (KO04975). For the age 27–28–30 group, upregulated genes were significantly enriched in 15 pathways (Figure 2D, Appendix A), while the downregulated genes were significantly enriched in 11 pathways (Figure 2F, Appendix A).

### 3.5. Time Series Analysis

According to the dynamic gene expression changes over time, four giant panda models were identified, and the trend of change of partial DEG expression with age was summarized through soft clustering (Figure 4A). The 630 genes in cluster 1 were highly expressed in 11-year-old giant pandas but were low in other age groups. The 1182 genes in cluster 2 were highly expressed in the young giant pandas, and the expression levels of these genes gradually decreased with an increase in age. The expression of the cluster 3 gene was higher in 3−, 4−, 5−, 18−, 19−, 21−, 22-, 23-, and 26-year-olds and lower in the other age groups, especially in 27–30-year-old groups of giant pandas. The 1228 genes in the cluster 4 expression pattern exhibited high levels of expression in the high age groups and low expression levels of expression in the younger groups.

Gene ontology (GO) (Figure 4B, Appendix A) and KEGG (Figure 4C, Appendix A) enrichment analysis were performed to assess the function of DEG with different patterns during giant panda development. The analyses revealed that the genes with high expression in cluster 1 at the age of 11 were mostly related to energy metabolism and ATP generation. They were also involved in the differentiation of Th1 and Th2 cells and in heat production. Cluster 2 genes were involved in gene expression, peptide biosynthesis, translation, and synthesis of various organelles. KEGG analysis showed that these genes were also involved in antigen processing and presentation, thermogenesis, and intestinal IgA production. Cluster 3 expression pattern genes were involved in regulating cell cycle and gene expression, as well as B cell receptor and T cell receptor signaling pathways. Cluster 4 genes were involved in the immune (innate immune) response, inflammatory response regulation, and Toll-like receptor signaling.

## 4. Discussion

In this study, many biological samples with a wide age distribution (3–30 years) and sufficient biological replicates were collected for scientific analysis. Differentially expressed genes based on age as a grouping basis were studied because previous studies have shown that age has a much more significant impact on gene expression patterns than gender [19]. Q < 0.05 was chosen as the threshold for all differences.

Genetic analysis of giant pandas of different age groups demonstrated no significant difference between the age 11 group and the age 3 group. GO enrichment analysis revealed that some upregulated genes such as CCL5, RSAD2, ACOD1, and STAT2 were involved in immune responses. CCL5 is a chemokine that is expressed initially in viral infection [20,21]. They are known to activate several cells directly involved in antiviral responses, including NK cells, T CD4+ lymphocytes, monocytes, mast cells, and dendritic cells [22]. RSAD2 is one of the few ISGs with direct antiviral activity. It also plays a vital role in regulating innate immunity [23]. STAT2, unlike others in the STAT family, is only involved in type I and III interferons (IFN-I/III) signaling pathways and also plays a role in innate immunity. Most of the genes involved in the immune response play essential roles in innate immune responses, indicating that the innate immunity of giant pandas is in continuous development during the growth. Additionally, CPT1A was highly expressed in the relatively young group. CPT1A is the first rate-limiting enzyme of fatty acid oxidation, and its high expression levels are conducive to fatty acid metabolism [24].

The number of differentially expressed genes between the age 27–28–30 group and the age 3 group was significantly different. Therefore, a hierarchical clustering of DEGs across seven samples was carried out. All the downregulated genes clustered in one group and all the upregulated genes clustered in the other. Similarly, the samples were also clustered into two groups by age (Figure 2B). DEGs enrichment analysis showed that these genes were mostly related to immune function. In particular, pandas in the age 27–28–30 group were observed to have upregulated expressions of genes responsible for innate immunity, including CCL5, TLR2, OAS2, and ISG15s, compared with the age 3 group (Table 2). On the other hand, the genes related to adaptive immunity were significantly downregulated, including those related to B cell and T cell differentiation (Table 3). An aspect of aging is the decline in the integrity of the immune system, leading to an increase in the body’s susceptibility to diseases [25]. Elderly giant pandas may compensate for the decline in adaptive immune function by improving their innate immunity. Further, the genes related to hematopoietic function were downregulated in older pandas. Interestingly, KEGG enrichment analysis of upregulated genes revealed a pathway associated with osteoclast differentiation, which has previously been reported to be associated with bone immunity and other bone diseases [26]. The downregulated genes were involved in the production of intestinal IgA, indicating that certain intestinal immune functions could be lacking in elderly giant pandas.

Giant pandas 6–20 years of age were considered adults. The gene expression analysis in the two selected age groups, i.e., age 11 and age 18–19, revealed many DEGs. Compared with age 11, 121 genes were upregulated, and 31 genes were downregulated in the age 18–19 group. GO enrichment analysis showed that these upregulated genes were significantly enriched in membrane-enclosed lumen (GO:0031974), organelle lumen (GO:0043233), protein binding (GO:0005515), and 94 other GO terms (Appendix A). Downregulated genes were not significantly enriched in any of the GO terms. The most significantly upregulated gene was HSP90AA1. This gene encodes a protein, Hsp90α, which has been reported to be associated with various diseases [27]. Other than that, there were no significant immune or metabolic changes in the adults.

In the aged giant panda group (21–30 years old), a comparative study of the change in expression levels of the genes with aging is crucial for preventing age-related diseases. The data of the 21–22 age group and the 27–28–30 age group were compared. It was observed that, in the age 27–28–30 group, 58 genes were upregulated, and 54 genes were downregulated. Enrichment analysis of DEGs revealed that the upregulated genes were significantly enriched in nine GO terms (Appendix A). However, the downregulated genes were not significantly enriched in any of the GO terms. GO term analysis of the upregulated genes showed that the upregulated genes, such as PLSCR1, CLEC7A, JUNB, and BATF3, were primarily involved in immune responses. In human studies, PLSCR1 has been found to play an essential role in IFN-dependent antiviral responses [28]. PLSCR1 directly interacts with and affects the function of several viral proteins, playing a pivotal role in PLSCR1-mediated antiviral activity [29,30,31]. The CLEC7A gene encodes dectin-1 and is widely expressed in the immune system. It is involved in biological processes such as allergies, cancer, autoimmune diseases, aseptic inflammation, and even aging [32]. JUNB mediates T cell development in the immune system [33]. BATF3 promotes the development of CD8α+ Conventional Dendritic cells (cDCs) by maintaining IRF8 and participating in the immune response [34]. MHC class II genes regulate Toll-like receptors, mediating innate immune responses in addition to their central role in adaptive immunity [35]. GO classification of downregulated genes was used to explain their functions. The analysis found that genes involved in immune response, such as CCL5, CCR9, and EPAS1, were downregulated. CCR9 is a G-protein-coupled receptor involved in biological processes such as immunity, inflammation, and tumors [36,37]. This result indicates that the elderly giant pandas enhanced the expression of some immune genes with age and compensated for the downregulated immune genes to resist the invasion of viruses, including innate immunity and adaptive immunity.

Time series analysis revealed that giant pandas had a high energy metabolism rate in the age 11 group. The genes strongly expressed at the age of 11 were enriched into the energy-related GO term. Protein misfolding is more likely to occur in high-salt environments, manifested in biological aging. ATP is an energy currency in animals, maintaining various metabolic networks of cells and also playing a pivotal role in protein stability through various mechanisms [38]. Almost all of the genes highly expressed in 11-year-old pandas are directly or indirectly related to ATP production. The analysis of cluster 2 genes revealed that giant pandas have a relatively vigorous metabolism when they are young. However, with age, the gene expression levels of adaptive immunity of the giant panda decrease. This decreased expression level of essential genes involved in antigen processing and presentation is bound to affect the integrity of the adaptive immune system. The downregulation of some of the genes engaged in heat production indicates that elderly pandas may be capable of generating less heat. The analysis of cluster 4 genes further proved that the expression of genes related to innate immunity is upregulated with aging in giant pandas. The genes related to regulating inflammatory response are also upregulated to a certain extent.

## 5. Conclusions

Giant pandas are known to undergo significant changes in immune-related genes with age. This study showed that congenital immunity developed faster in giant pandas as they grew from juvenility to adulthood. This was consistent with the results of a previous study showing that the immune system of giant pandas gradually improved from infancy to adulthood [39]. Our study found that innate immunity might develop faster than adaptive immunity. There was also an upregulation in innate immunity and downregulation in adaptive immunity in old giant pandas relative to the young. Additionally, previous studies have not proven the humoral immunity and change in T cell immunity [40], these factors supplementing the prevention of aging-related diseases. The immune system of the elderly giant panda is still changing. In the process of aging, the body undergoes the adjustment of some immune genes to compensate for the loss of other immune functions. It was also shown that giant pandas at the age of 11 promoted fat consumption to meet the storage of liver glycogen and high ATP production. Further studies are required to understand the biological significance of these processes.

## Figures and Tables

**Figure 1 vetsci-09-00667-f001:**
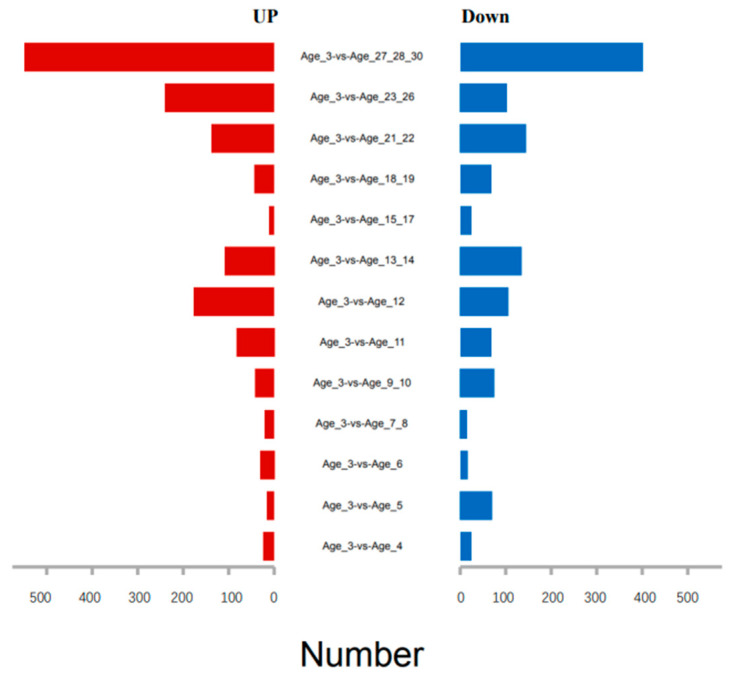
The number of differentially expressed genes of giant pandas in different age groups compared to three-year-old groups.

**Figure 2 vetsci-09-00667-f002:**
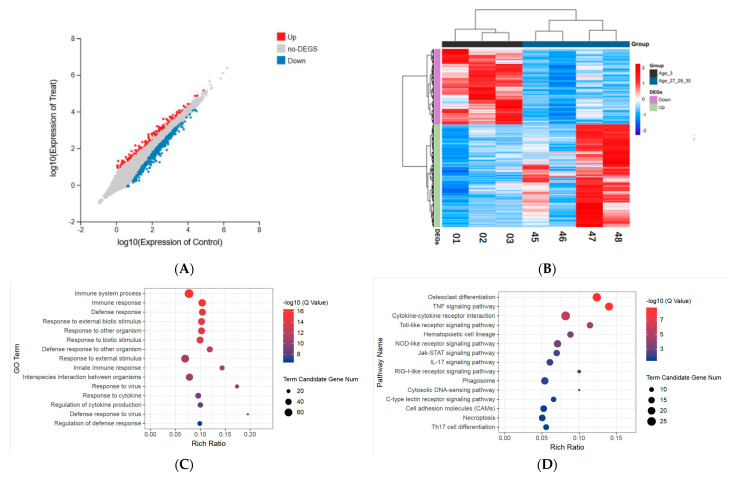
Differentially expressed genes in the age 27−28−30 group compared to the age 3 group. (**A**) Scatter plot. X and Y axes represent log values of gene expression levels. Red represents upregulated DEG, blue represents downregulated DEG, and gray represents non−DEG. (**B**) Hierarchical clustering method was used to draw the heat map of DEGs. Displays expression values for two groups of seven individuals. Each column represents a sample, and each row represents a gene. Red indicates the gene is upregulated, and blue indicates the gene is downregulated. (**C**) GO enrichment analysis of upregulated DEGs. (**D**) KEGG pathway enrichment analysis of upregulated DEGs. (**E**) GO enrichment analysis of down-regulated DEGs. (**F**) KEGG pathway enrichment analysis of down-regulated DEGs.

**Figure 3 vetsci-09-00667-f003:**
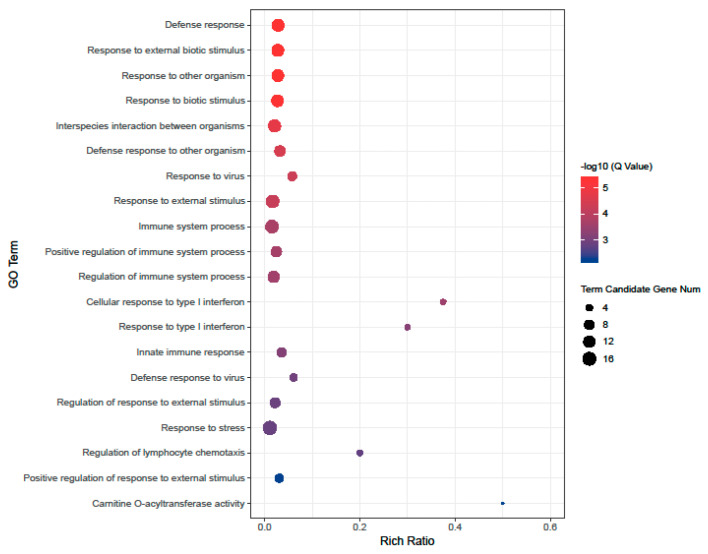
GO enrichment analysis of upregulated DEGs in the age 11 group.

**Figure 4 vetsci-09-00667-f004:**
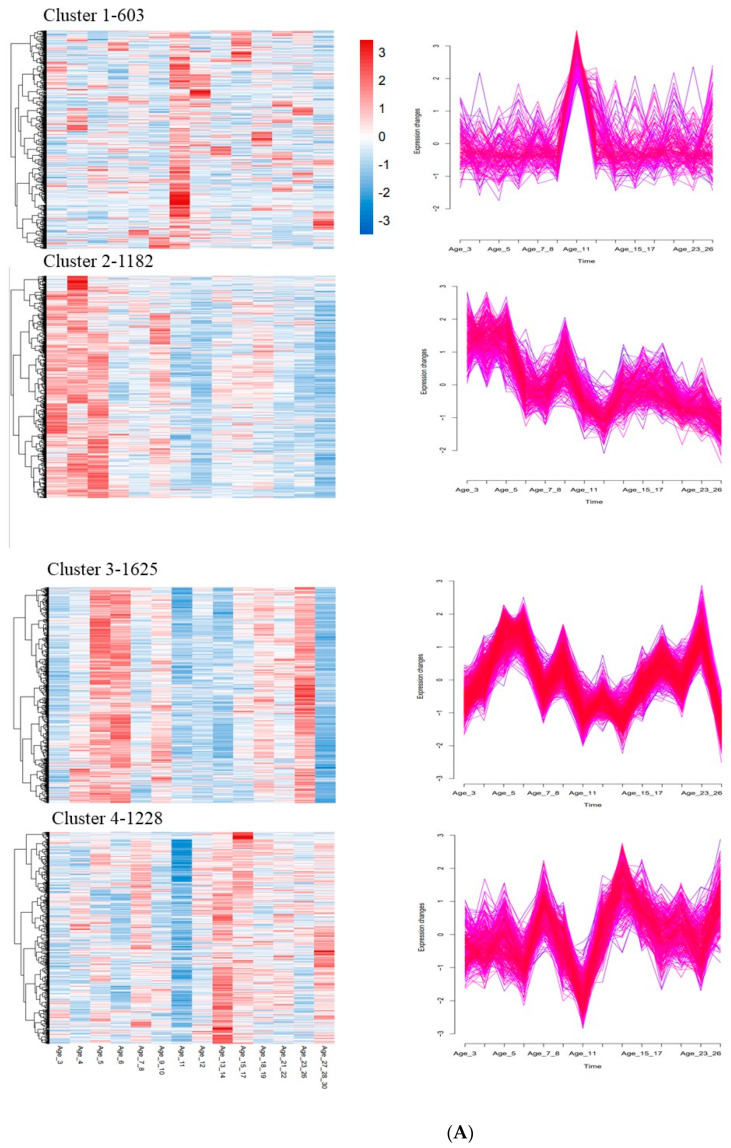
Dynamic time series expression patterns and pathway analysis. (**A**) Heatmap of four clusters of differentially expressed genes identified by soft clustering representing different gene expression patterns across the different time points as indicated. Gene Ontology (GO) (**B**) and Kyoto Encyclopedia of Genes and Genomes (KEGG) (**C**) enrichment analyses for the genes in the four clusters.

**Table 1 vetsci-09-00667-t001:** Summary of sequencing and assembly of the transcriptome.

Sample	Age	Gender	Group	Total Raw Reads (M)	Total Clean Reads (M)	Total Clean Bases (Gb)	Clean Reads Q30 (%)	Clean Reads Ratio (%)	Total Mapping (%)
01	3	Male	Age_3	43.82	42.97	6.45	93.11	98.05	92.69
02	3	Female	Age_3	43.82	42.91	6.44	92.94	97.92	91.08
03	3	Female	Age_3	43.82	43.09	6.46	93.55	98.34	92.13
04	4	Male	Age_4	43.82	42.96	6.44	93.63	98.04	91.93
05	4	Female	Age_4	43.82	42.86	6.43	93.73	97.8	91.91
06	4	Male	Age_4	43.82	42.86	6.43	93.56	97.8	91.62
07	5	Female	Age_5	43.82	42.87	6.43	92.84	97.83	90.83
08	5	Female	Age_5	43.82	43.09	6.46	92.78	98.32	89.88
09	5	Female	Age_5	45.57	43.58	6.54	89.68	95.63	86.82
10	6	Female	Age_6	43.82	42.89	6.43	92.75	97.88	91.05
11	6	Female	Age_6	43.82	42.99	6.45	92.58	98.11	91.71
12	6	Male	Age_6	43.82	42.44	6.37	89.63	96.86	85.84
13	6	Female	Age_6	43.82	42.11	6.32	89.45	96.1	86.55
14	7	Male	Age_7_8	43.82	42.62	6.39	92.59	97.25	91.56
15	7	Male	Age_7_8	43.82	42.6	6.39	93.09	97.21	91.08
16	8	Male	Age_7_8	43.82	42.91	6.44	92.89	97.91	91.24
17	8	Male	Age_7_8	43.82	42.95	6.44	92.87	98.01	91.26
18	8	Female	Age_7_8	43.82	42.89	6.43	93.33	97.87	92.38
19	9	Female	Age_9_10	43.82	43.11	6.47	92.5	98.39	91.93
20	10	Female	Age_9_10	43.82	43.04	6.46	93.48	98.22	92.35
21	10	Female	Age_9_10	43.82	42.63	6.39	93.5	97.28	92.16
22	11	Female	Age_11	43.82	42.91	6.44	93.3	97.93	90.86
23	11	Female	Age_11	43.82	43.07	6.46	93.81	98.28	91.9
24	11	Female	Age_11	43.82	43.11	6.47	92.94	98.37	92.51
25	12	Male	Age_12	43.82	42.72	6.41	93.11	97.48	89.86
26	12	Male	Age_12	43.82	43.04	6.46	93.39	98.23	91.8
27	12	Female	Age_12	43.82	43.07	6.46	93.16	98.29	91.63
28	13	Female	Age_13_14	43.82	42.15	6.32	93.54	96.19	90.35
29	13	Male	Age_13_14	43.82	42.99	6.45	93.34	98.1	91.47
30	14	Female	Age_13_14	43.82	43.32	6.5	92.79	98.87	91.19
31	14	Male	Age_13_14	43.82	43.26	6.49	93.33	98.71	91.69
32	15	Female	Age_15_17	43.82	43.09	6.46	93.03	98.32	92.25
33	15	Male	Age_15_17	43.82	43.14	6.47	93.19	98.45	91.73
34	17	Female	Age_15_17	43.82	42.99	6.45	93.69	98.11	92.44
35	18	Female	Age_18_19	43.82	43.06	6.46	93.24	98.26	92.34
36	18	Female	Age_18_19	43.82	43.11	6.47	93.2	98.38	91.62
37	19	Female	Age_18_19	43.82	43.1	6.47	93.39	98.36	91.25
38	19	Male	Age_18_19	43.82	43	6.45	93.34	98.14	92.29
39	21	Female	Age_21_22	43.82	42.89	6.43	93.11	97.87	91.46
40	22	Female	Age_21_22	43.82	43.12	6.47	93.13	98.4	91.53
41	22	Female	Age_21_22	43.82	42.91	6.44	93.16	97.93	91.88
42	23	Male	Age_23_26	43.82	43.01	6.45	92.85	98.14	90.03
43	26	Female	Age_23_26	43.82	42.65	6.4	92.97	97.33	91.51
44	23	Female	Age_23_26	45.57	43.56	6.53	89.28	95.58	85.69
45	27	Female	Age_27_28_30	43.82	43.03	6.45	92.73	98.18	90.75
46	28	Male	Age_27_28_30	43.82	42.97	6.45	93.31	98.06	90.83
47	30	Female	Age_27_28_30	43.82	42.93	6.44	93.32	97.97	91.54
48	30	Female	Age_27_28_30	43.82	42.72	6.41	92.99	97.48	90.06

**Table 2 vetsci-09-00667-t002:** Upregulated innate immune genes in the elderly giant pandas.

Gene ID	Gene symbol	log2FC	Q value
100463753	CCL5	1.197677414	0.006639364
100465129	TLR2	0.672103431	0.045445131
100465831	LOC100465831	1.393709166	0.032208904
100466958	TMCO6	0.625425843	0.044165086
100467144	TLR8	0.772216276	0.028726888
100467227	SLPI	1.084231605	0.040962187
100468112	OAS2	1.901277089	0.00136348
100469387	ISG15	2.542014757	0.009571346
100469482	NOD2	1.392587422	0.003516809
100472167	STAT2	1.657003098	0.001891366
100474026	STAT1	1.162051454	0.040464308
100475582	LOC100475582	0.769064873	0.044459423
100475654	TLR5	1.265140332	0.007245626
100475782	LOC100475782	3.167993087	6.07 × 10^−7^
100476031	TRIM8	1.217417982	0.00289373
100477088	MX1	2.412068631	0.018689447
100477434	LOC100477434	0.813886584	0.005457552
100477995	RAB20	2.011557998	6.51 × 10^−5^
100481969	IRF1	1.233846742	0.000175493
100482061	SLC11A1	1.498952435	2.74 × 10^−5^
100483325	LOC100483325	2.131324274	0.01898041
105238896	LOC105238896	1.108664282	0.041700258
109488883	LOC109488883	1.392799023	0.000629463
BGI_novel_G000507	BGI_novel_G000507	1.074982321	0.043726236
BGI_novel_G000508	BGI_novel_G000508	0.802736093	0.013117263
BGI_novel_G000816	BGI_novel_G000816	1.80240837	6.94 × 10^−5^
BGI_novel_G000830	BGI_novel_G000830	2.064624645	0.000209942
BGI_novel_G000834	BGI_novel_G000834	2.921364095	3.50 × 10^−12^

**Table 3 vetsci-09-00667-t003:** Downregulated immune genes in older adults.

Gene ID	Gene symbol	log2FC	Q value
100463670	SYNCRIP	−0.657043606	0.022428329
100463973	SATB1	−1.087420722	0.001571307
100464341	ILF2	−0.599904586	0.03021565
100464393	LOC100464393	−1.100916666	0.000807226
100465982	LEF1	−1.200422244	0.000864714
100467178	BACH2	−1.407743785	0.015084227
100467628	LTB	−0.84441179	0.004005136
100469077	LAX1	−0.942370345	0.044459423
100469792	RHOH	−0.883131268	0.036375059
100471131	CD79B	−1.964581188	9.70 × 10^−7^
100471141	CD19	−1.433097254	0.00042723
100471593	BCL11B	−1.084197888	0.020106663
100471626	CD79A	−1.163510008	0.006749405
100472031	LOC100472031	−0.865494399	0.010478855
100475265	WNT10B	−1.55344294	0.028829421
100475502	CD7	−0.815269322	0.00519229
100476508	MS4A1	−1.032444512	0.001300313
100476528	TCF7	−0.861882414	0.032397565
100476544	KLHL6	−1.037565296	0.00887367
100477968	STON2	−1.409667727	0.024177379
100478290	LOC100478290	−1.161973596	0.01488409
100478982	IMPDH2	−0.777026522	0.008127539
100479355	BLK	−1.452355267	0.008778106
100479879	FCRL1	−1.998071764	3.56 × 10^−5^
100480638	ATM	−1.481610202	6.80 × 10^−8^
100482026	SOX4	−2.850078573	3.40 × 10^−10^
100482462	CD3E	−0.758174358	0.038685191
100482465	CR2	−1.415727356	0.003516809
105240103	LOC105240103	−2.821011447	2.31 × 10^−8^

## Data Availability

All sequencing data have been submitted to the NCBI database (http://www.ncbi.nlm.nih.gov/bioproject/893394, accessed on 23 October 2023).

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
