# Peer review of "Using Blood Transcriptome Analysis to Determine the Changes in Immunity and Metabolism of Giant Pandas with Age"

_vetsci, 2022, doi:10.3390/vetsci9120667_

Round 1
Reviewer 1 Report
Dear authors,
This work tries to analyze the difference in gene expression related to ageing in giant panda by RNA-seq methodology. The manuscript presents several mistakes of format, as number of references, among others. Furthermore, the authors use 48 samples, a low number, and they do not explain how many are to different age, and to different sex. Only they explain that they use 16 male and 33 females, a number very unbalanced. These thinks difficult to review and could be the results could have as a consequence a bias in the results obtained, since I cannot know if the differences found are due to age, sex, or other conditions not explained by the authors. For all these reasons, I reject the paper for publication.
Author Response
Dear Editor and Reviewers:
On behalf of my co-authors, we are very grateful to you for giving us an opportunity to revise our manuscript. we appreciate you very much for your positive and constructive comments and suggestions on our manuscript entitled “Using Blood Transcriptome Analysis to Determine the Changes in Immunity and Metabolism of Giant Pandas with Age" (Manuscript ID:vetsci-1938342).
We have studied reviewers' comments carefully and tried our best to revise our manuscript according to the comments. The following are the responses and revisions I have made in response to the reviewers' questions and suggestions on an item-by-item basis. Thanks again to the hard work of the editor and reviewer!
Comment: except for one article cited in this study (Reference xxxviii), another important literature (Huang et al., The immune and metabolic changes with age in giant panda blood by combined transcriptome and DNA methylation analysis, aging, 2020, 12(21): 21777-97) was not included in the discussion or introduction part of this manuscript. The two published paper were also used blood RNa-SEQ to identify the immune-related genes. So, atuhors should provided more details about the difference in these related papers
Response: Thank you for your suggestions. We have revised the format of the references. In this study, all the big bears were healthy individuals. Previous studies have shown that the effect of gender on transcriptome expression differences is far less than that of age. Therefore, we have not explored the differences caused by gender in this study. I am sorry that this experimental design has not been affirmed by you.
Thank you again for your questions and suggestions.
Wish you a happy life!
sincerely
song liu
November 6, 2022

Reviewer 2 Report
This manuscript try to reveal the differentially expressed genes and pathways with age by RNA-seq of blood tissues. The experiment designed well and the results will helpful to understand the disease prevention and energy metabolsim of giant panda. some major concerns and modifications are needed before acepted for publication.
1. except for one article cited in this study (Reference xxxviii), another important literature (Huang et al., The immune and metabolic changes with age in giant panda blood by combined transcriptome and DNA methylation analysis, aging, 2020, 12(21): 21777-97) was not included in the discussion or introduction part of this manuscript. The two published paper were also used blood RNa-SEQ to identify the immune-related genes. So, atuhors should provided more details about the difference in these related papers.
2. In the second paragragh of page 2, authors pointed out Baylisascaris Schroederi and canine distemper virus (CDV) killed half of the panda population between 2001-2005. Please provide more references? Meanwhile, some DEGs in this study are involved in this virus prevention?
3. In page 4, there are some wrong that all pandas were divided into four groups, young(3 years), mature (11 years), middle-age(18-19 years)and old (27-30 years). In fact, female/male giant pandas reach sexual maturity at about 4-6 years, and pandas aged 18-19 years are also old. Therefore, the classification of age is not suitable and resonable here.
4. more details of materials and methods are needed.
minor issues:
5. In page 3, the acknowledgments should be placed at the end of this manscript.
6. please deleted the wrong Chines word in the legend of figure1.
7. Figure 2, age groups 11 should be "age 11 group".
8. the functions of listed gene in table 3 can be provided.
9. all figure's solution are low.
Author Response
Dear Editor and Reviewers:
On behalf of my co-authors, we are very grateful to you for giving us an opportunity to revise our manuscript. we appreciate you very much for your positive and constructive comments and suggestions on our manuscript entitled “Using Blood Transcriptome Analysis to Determine the Changes in Immunity and Metabolism of Giant Pandas with Age" (Manuscript ID:vetsci-1938342).
We have studied reviewers' comments carefully and tried our best to revise our manuscript according to the comments. The following are the responses and revisions I have made in response to the reviewers' questions and suggestions on an item-by-item basis. Thanks again to the hard work of the editor and reviewer!
Comment: except for one article cited in this study (Reference xxxviii), another important literature (Huang et al., The immune and metabolic changes with age in giant panda blood by combined transcriptome and DNA methylation analysis, aging, 2020, 12(21): 21777-97) was not included in the discussion or introduction part of this manuscript. The two published paper were also used blood RNa-SEQ to identify the immune-related genes. So, atuhors should provided more details about the difference in these related papers
Response: Thank you for your reminding. We have added the reference "Huang et al., The immune and metabolic changes with age in giant panda blood by combined transcription and DNA methylation analysis, aging, 2020, 12 (21): 21777-97" to the reference and discussed the differences between the reference and this study. The reference focuses on the regulatory relationship between gene methylation and gene expression, At the same time, the changes of immunity with age are "the risk of cancer of giant pandas changes with age", "the innate immune system of giant pandas gradually improves with the development from infancy to adulthood" and "the inhibition regulation of B cells gradually weakens during the adult process of giant pandas". Our research results found that the elderly giant pandas have both the upregulation of innate immunity and the downregulation of adaptive immunity compared with the young giant pandas, and the genes involved in innate immunity and adaptive immunity in the elderly giant pandas are also changing with the aging process.
Comment No. 2: In the second paragragh of page 2, authors pointed out Baylisascaris Schroederi and canine distemper virus (CDV) killed half of the panda population between 2001-2005. Please provide more references? Meanwhile, some DEGs in this study are involved in this virus prevention?
Response: We are very sorry that our negligence caused problems in literature citation. We have applied the literature "8. Xie, Y.; Zhou, X.; Zhang, Z.; Wang, C.; Sun, Y.; Liu, T.; Gu, X.; Wang, T.; Peng, X.; Yang, G Absence of genetic structure in Baylisascaris schroederi populations, a giant panda parasite, determined by mitochondrial sequencing. Parasit Vectors 2014, 7, 606-606, doi: 10.1186/s13071-014-0606-3. "and" 10. Feng N, Yu Y, Wang T, Wilker P, Wang J, Li Y, Sun Z, Gao Y, Xia X. Fatal Canine dispatcher virus effect of giant panda in China. Sci Rep 2016, 6, 27518, doi: 10.1038/srep27518. "as supplements.
In this study, some DEGs such as CCL5, RSAD2 and genes involved in B cell activation and differentiation play a key role in antiviral response.
Comment No. 3: In page 4, there are some wrong that all pandas were divided into four groups, young(3 years), mature (11 years), middle-age(18-19 years)and old (27-30 years). In fact, female/male giant pandas reach sexual maturity at about 4-6 years, and pandas aged 18-19 years are also old. Therefore, the classification of age is not suitable and resonable here.
Response: Thank you for your suggestion. We directly group the giant pandas by age. Now we change it to "age 3group", "age 11 group", "age 18-19 group", "age 27-28-30 group".
Comment No. 4: more details of materials and methods are needed.
Response: We have added more descriptions of materials and methods in the first half of the article.
Comment No. 5: In page 3, the acknowledgments should be placed at the end of this manscript.
Response: Thank you for your suggestion. We have moved acknowledgments to the end of the manuscript.
Comment No. 6: please deleted the wrong Chines word in the legend of figure1.
Response: We are very sorry that this problem was caused by our negligence, and we have corrected it.
Comment No. 7: Figure 2, age groups 11 should be "age 11 group".
Response: Thank you for your suggestion. We have corrected it.
Comment No. 8: the functions of listed gene in table 3 can be provided.
Response: Thank you for your suggestion. In this study, we have put these genes into GO and KEGG enrichment analysis, and we have not done detailed research on the specific function of a single gene. We are very sorry for this.
Comment No. 9: all figure's solution are low
Response: Thank you for your suggestion. We have tried to improve the resolution and aesthetics of the image, and will provide the original image.
Thank you again for your questions and suggestions.
Wish you a happy life!
sincerely
song liu
November 6, 2022

Round 2
Reviewer 1 Report
The authors have corrected the previous version, and I consider that the article can be published.
Author Response
Dear reviewers,
We are very glad that you recognized our manuscript and thank you for reviewing it again.
I wish you all the best!
Kind regards,
song liu
Reviewer 2 Report
The authors revised carefully and respond well to my concerns. Although some details of materials and methods are provided by authors, section '2.3 Library preparation and sequencing' was lack of specific procedures rather than simply described the protocol. So, I satisfied the current format of manuscript and can be accepted for publication after above mentioned minor modifications.
Author Response
Dear reviewers,
Thank you very much for reviewing our manuscript again. I'm sorry for replying to you so late, because we are trying to solve the problems in the manuscript. However, it is a pity that after many exchanges with the company, we can only get the current detailed experimental steps. We are very sorry, but we are still trying to get the detailed steps of database building.
Thanks again for your review.
Kind regards,
song liu